# Dissecting cascade computational components in spiking neural networks

**Shanshan Jia**[1], **Dajun Xing**[2], **Zhaofei Yu**[1]\*, **Jian K. Liu**[3]\*

**1** Institute for Artificial Intelligence, Department of Computer Science and Technology, Peking University, Beijing, China, **2** State Key Laboratory of Cognitive Neuroscience and Learning, IDG/McGovern Institute for Brain Research, Beijing Normal University, Beijing, China, **3** School of Computing, University of Leeds, Leeds, United Kingdom

\* yuzf12@pku.edu.cn (ZY); j.liu9@leeds.ac.uk (JKL)

## Abstract

Finding out the physical structure of neuronal circuits that governs neuronal responses is an important goal for brain research. With fast advances for large-scale recording techniques, identification of a neuronal circuit with multiple neurons and stages or layers becomes possible and highly demanding. Although methods for mapping the connection structure of circuits have been greatly developed in recent years, they are mostly limited to simple scenarios of a few neurons in a pairwise fashion; and dissecting dynamical circuits, particularly mapping out a complete functional circuit that converges to a single neuron, is still a challenging question. Here, we show that a recent method, termed spike-triggered non-negative matrix factorization (STNMF), can address these issues. By simulating different scenarios of spiking neural networks with various connections between neurons and stages, we demonstrate that STNMF is a persuasive method to dissect functional connections within a circuit. Using spiking activities recorded at neurons of the output layer, STNMF can obtain a complete circuit consisting of all cascade computational components of presynaptic neurons, as well as their spiking activities. For simulated simple and complex cells of the primary visual cortex, STNMF allows us to dissect the pathway of visual computation. Taken together, these results suggest that STNMF could provide a useful approach for investigating neuronal systems leveraging recorded functional neuronal activity.

## Author summary

It is well known that the computation of neuronal circuits is carried out through the staged and cascade structure of different types of neurons. Nevertheless, the information, particularly sensory information, is processed in a network primarily with feedforward connections through different pathways. A peculiar example is the early visual system, where light is transcoded by the retinal cells, routed by the lateral geniculate nucleus, and reached the primary visual cortex. One meticulous interest in recent years is to map out these physical structures of neuronal pathways. However, most methods so far are limited to taking snapshots of a static view of connections between neurons. It remains unclear how to obtain a functional and dynamical neuronal circuit beyond the simple scenarios of

**Data Availability Statement:** The code used to generate the results in this paper is available on https://github.com/jiankliu/STNMF-SNN.

**Funding:** This work was supported by National Natural Science Foundation of China Grants

62176003, 61961130392 and 62088102 (ZY) and
32171033 (DX), and Royal Society Newton
Advanced Fellowship of UK Grant NAF-R1-191082
(JKL). The funders had no role in study design,
data collection and analysis, decision to publish, or
preparation of the manuscript.

a few randomly sampled neurons. Using simulated spiking neural networks of visual pathways with different scenarios of multiple stages, mixed cell types, and natural image stimuli, we demonstrate that a recent computational tool, named spike-triggered nonnegative matrix factorization, can resolve these issues. It enables us to recover the entire structural components of neural networks underlying the computation, together with the functional components of each individual neuron. Utilizing it for complex cells of the primary visual cortex allows us to reveal every underpinning of the nonlinear computation. Our results, together with other recent experimental and computational efforts, show that it is possible to systematically dissect neural circuitry with detailed structural and functional components.

## Introduction

One of the cornerstones for developing novel algorithms of neural computation is to utilize different neuronal network structures extracted from experimental data. The connectome, *wiring diagrams*, becomes an increasingly important topic, especially, for those relatively simple neuronal circuits that are well-studied, such as the retina [1–6]. Based on certain experimental techniques, the wiring diagram of neuronal connections has been identified for simple animal models, including Caenorhabditis elegans [7], Drosophila [8], and tadpole larva [9]. So far, most of these methods can only take a static view of connection strengths for neural circuits by imaging data, and the dynamics of synaptic strengths, which is a unique and essential feature of neural computation, is hardly estimated.

The function of neuronal computation has been shown to be highly dynamics in the temporal domain with strong adaptation to stimulus statistics [10, 11], nonlinear temporal integration [12, 13], trial specific temporal dynamics [14, 15]. The question of how to obtain a functional and dynamical neuronal circuit has been studied experimentally [16] and computationally [17, 18] with great efforts in recent years. Spike-triggered non-negative matrix factorization (STNMF) is one of the methods proposed to infer the underlying structural components of the retina based on temporal sequences of spiking activities recorded in ganglion cells [17]. STNMF takes the advantage of machine learning technique NMF, which has a great capacity to capture local structures of given datasets [19]. It has been used recently to identify functional units localized in space and time in neuronal activities [20–26]. STNMF takes a step further to analyze the mapping between stimuli and neural responses leveraging neural spikes while leaving out non-responsive stimuli [17, 27], with an assist of spare coding, as neurons generally fire with a low rate of spikes [28].

However, it is not clear whether the STNMF is applicable to dissecting a complete neural circuit with multiple stages or layers all formed by multiple spiking neurons. Here we address this question by comparing the true dynamic connection and strengths in a model and those estimated by STNMF. The model is a spiking neural network mimicking the feedforward connection at multiple stages in early visual systems, including the retinal ganglion cells (RGCs), lateral geniculate nucleus (LGN), and primary visual cortex (V1). We first demonstrated STNMF can reliably infer presynaptic spikes from postsynaptic spikes and obtain presynaptic strengths and dynamics for multiple spiking neurons projecting to a single postsynaptic neuron. Then we showed that when there are more than one postsynaptic neurons, STNMF is able to map out the entire neural circuit by analyzing each individual postsynaptic neuron. With a multiple layer neural network, STNMF can identify each layer in the model. Particularly, STNMF is applicable to the complex stimulus of natural images. Finally, we show that

STNMF is applicable to V1-like simple and complex cells of neural networks with mixed cell types. Taken together, our results indicate that STNMF is an effective approach to describe the underlying neural circuits using neural spikes of single cells.

## Methods

### Neural network model

We simulated a simple version of the early visual pathway, from the retina ganglion cells (RGCs) to the lateral geniculate nucleus (LGN) and primary visual cortex (V1), by feedforward layered neural networks with spiking neurons, under different scenarios of network connections.

We first employed a two-layer spiking neural network to illustrate the workflow of STNMF. There are four presynaptic RGCs in the first layer, where each RGC was modeled as a linear-nonlinear Poisson spiking neuron [29] with an OFF type spatiotemporal receptive field filter $k$, consisting 2x2 pixels in space and a biphasic temporal filter, together with a nonlinearity $f$, as its specific computational components. The input stimulus $s(t)$ was given by a sequence of random binary black-white checkers with 8x8 pixels typically used for visual neuroscience experimentalists to map the receptive field of neurons. The nonlinearity $f$ is expressed as $f(x) = x$ if $x >= 0$, $f(x) = 0$ if $x < 0$. The model output was the firing probability $r = f(k * s(t))$, where $*$ represents spatiotemporal convolution. A sequence of spikes was generated by the Poisson process. Each presynaptic neuron has a different spatial filter where the focus is located at different parts of images. As a result, output spike trains are different between presynaptic neurons. The spiking output of each neuron was sent out to one postsynaptic neuron in the second layer with specific synaptic weights.

In the second layer, a postsynaptic LGN neuron was modeled by a leaky integrate-and-fire neuron as $\tau_m \, dV/dt = -(V(t) - V_{rest}) + RI(t) + V_{noise}$ where $V(t)$ is the membrane potential at time $t$, $V_{rest}$ as the resting potential, $\tau_m = 10$ ms as the membrane time constant. $R = 1$ is the finite leak resistance. $I$ represents the postsynaptic current received by the neuron. $V_{noise}$ represents the noise that obeyed the normal distribution with a mean value of 0 and a standard deviation of 0.02 mV. The LGN neuron collected information from all the RGCs in the first layer by synaptic connections, such that its postsynaptic current $I = \sum_i w_i \sum_j I_{syn}(t - t_i^j)$, where the neuron $i$ is one of the RGCs, $w_i$ is the synaptic weight from the RGC $i$ to the LGN, and $I_{syn}$ is the synaptic current when there is a spike $j$ occurring at $t_i^j$. For simplicity, $I_{syn}$ was modeled as an alpha function as $I_{syn} = A \, \exp(-(t - t_i^j)/\tau_0)\Theta(t - t_i^j)$, where $A = 1$, $\tau_0 = 10$ ms and $\Theta(x)$ is the Heaviside step function. When the accumulated membrane potential achieved the threshold, the LGN neuron fired a spike. This network can be considered as a minimal model of a LGN consisting of four presynaptic RGCs.

We then extended this two-layer network to include multiple neurons and layers. We first included two LGNs in layer 2. We then considered a three-layer network, where there were six RGCs in layer 1, two LGN neurons in layer 2 and one V1 neuron in layer 3. We also examined a four-layer network. For the network model with mixed ON and OFF cells, we fixed temporal filters as negative, while adopting different polarities of spatial filter to indicate the sign of ON or OFF cells, such that OFF cells have positive spatial filters, while ON cells have negative spatial filters. As a result, the spatiotemporal filter as a multiplication of spatial and temporal filters is positive for ON cells and negative for OFF cells.

In all network models, neural models for RGCs, LGNs, and V1 were the same as above, e.g., neurons in layer 1 were modeled as linear-nonlinear Poisson neurons, while neurons in layer 2 and 3 were modeled as integrate-and-fire neurons. Spatial receptive fields of RGCs have

different locations on stimulus images. For simplicity, synaptic weights were fixed as 1, except for those specific cases with the mentioned values.

## Spike-triggered non-negative matrix factorization analysis

The STNMF method is inspired by a simple and useful method of system identification in visual neuroscience, named spike-triggered average (STA) [29], which uses every response spike to reversely correlate input stimuli. Briefly, for a spike $r^i$ occurring at time $t_i$, one can collect a segment of stimuli $s(\tau)^i = s(t_i - \tau)$ that precede that spike, where the lag $\tau$ denotes the timescale of history, into an ensemble of spike-triggered stimuli $\{s(\tau)^i\}$, then averages it overall spikes to get the STA filter $k(\tau) = \langle s(\tau)^i \rangle_i$. When stimuli are spatiotemporal white noise, the 3D STA filter can be decomposed by singular value decomposition to obtain the temporal filter and spatial receptive field [30].

The STNMF analysis was introduced in [17] and extended in [27]. Briefly, to reduce computational cost, we first applied pre-processing for the spike-triggered stimulus ensemble: for the $i$-th spike, the corresponding stimulus segment $s(\tau)^i$ is weighted averaged by temporal STA filter $k_t$: $\bar{s}^i = \sum_\tau s(\tau)^i \cdot k_t(\tau)$, such that time dimension $\tau$ is collapsed to a single frame of stimulus image for the $i$-th spike, termed effective stimulus image $\bar{s}^i$. With the ensemble of effective stimulus images $S = \{\bar{s}^i\}$ for all spikes, one can apply a semi-NMF algorithm [31], similarly to the analysis of a set of face images [19].

Specifically, the ensemble of effective stimulus images $S = \{\bar{s}^i\}$ can be rewritten as $S = (s_{ij})$, a $N \times P$ matrix with indexes $i = 1, \cdots, N$ for all $N$ spikes, and $j = 1, \cdots, P$ for all the pixels in $P$ images. NMF allows us to decompose the matrix as $S \approx WM$, where weight matrix $W$ is $N \times K$, module matrix $M$ is $K \times P$, and $K$ is the number of modules. Both stimuli $S$ and weights $W$ can be negative, but modules $M$ are still non-negative. The function to be minimized is $F = \| S - WM \|_F^2 + \lambda \sum_{j=1}^P \| M_j \|_1^2$, where the sparsity parameter $\lambda = 0.1$, and $\|v\|_1$ is the $L_1$ norm of a vector $v$. $\|.\|_F$ denotes the Frobenius norm of a matrix. The sparsity constraint here is to control the overall contribution to each spike from a set of modules in each column of $M$, rather than directly control the size of the receptive field. One can implement the minimization of $F$ as an alternating optimization of $W$ and $M$ based on the NMF toolbox [32]. The result of STNMF decomposition is a set of modules corresponding to spatial receptive fields of neurons, and one single weight matrix including information of synaptic wights and presynaptic spikes.

## Inferring presynaptic spikes

The STNMF weight matrix is specific to each individual spike of postsynaptic neurons. Thus, one can reconstruct all the possible spikes contributed by each presynaptic neuron [27]. In the two-layer model, LGN spikes were represented by incoming four RGCs, thus, each spike of LGN could be contributed by one of the RGCs. Inspired by the clustering viewpoint of STNMF, STNMF can classify all the LGN spikes into four subsets of spikes such that each subset of spikes is mainly, if not completely, contributed by one specific RGC. As each row corresponds to one individual spike, every spike can be classified according to the weight value of the STNMF matrix. For the model with OFF cells, and since modules are always non-negative, one can take the minimal value per row in weight matrix $w_{ij}$, for instance, $\min(w_{1k}) = \min_j(w_{1j})$ for the first row and first spike. The index $j$ indicates which presynaptic RGC should be for this specific spike. For ON cells, the maximal values were used to obtain the ON spikes. After lopping all rows/spikes, we can obtain a set of spikes belonging to a specific presynaptic RGC.

For the single LGN model, we obtained four subsets of spikes for four RGCs respectively. For the two-layer model with two LGNs, we have six subsets of spikes for six RGCs. For the three-layer model with two LGNs, we extracted six subsets of spikes for six RGCs in layer 1. To obtain spikes for each neuron in the middle layer, we pooled the spikes of RGCs into two pools, such that each pool collects four of six RGCs corresponding to one of the LGNs.

For the model with mixed ON and OFF cells, a similar approach was used. Instead of finding all minimal values, we computed both the minimum and maximum of each spike, then compared the absolute values of these two and used the maximal one to indicate the final index for presynaptic cells, e.g., if the absolute value of minimum is larger, the spike is from an OFF cell, otherwise, it is from an ON cell. In this way, all spikes can be attributed to either ON and OFF cells in the model. Otherwise, one can collect both sets of minimums and maximums as spikes to take into account the noise effect of neurons. Both approaches are applicable to extract spikes of upstream neurons.

Similarly, each individual element $w_{ij}$ is also the strength between the $i$-th postsynaptic spike and the presynaptic cell $j$. By averaging each column of the weight matrix, one can obtain a single weight value for each synaptic connection from presynaptic to postsynaptic cells.

## Mutual information carried by spikes

In order to characterize the quality of spikes inferred by STNMF, we computed mutual information (MI) carried by spikes for a given stimulus. Instead of Pearson correlation coefficients between two spike trains, MI is to quantify how much information is carried by spikes. We employed a previous approach to compute MI [11]. For a given spike train $spk_i$, the MI was computed as $\text{MI}_k(spk_i) = \int ds P(s_k|spk_i) log_2(P(s_k|spk_i)/P(s_k))$. In our model, each presynaptic neuron has the given spatiotemporal filter $k$ to convolve stimulus, then we name the convolved stimulus signal as $s_k$, which is also the project of the stimulus along with the direction of the filter. $P(s_k)$ is the probability distribution of the prior stimulus set along the filter $k$ direction, and $P(s_k|spk_i)$ is the probability distribution of spike-triggered stimuli in this direction given the spike train $spk_i$. The integral was evaluated by discretizing the convolved stimulus values $s$ with a bin size as 0.1 of the stimulus standard deviation. All information values were corrected for bias due to finite sampling following previous studies by using subsections of the data (80%-100%) and linear extrapolation to estimate the information value at infinite sample size [33, 34]. In our model, there are a number of presynaptic neurons with different filters. We have the corresponding spike trains generated by the model and inferred by STNMF. Thus, we can compute the MI between a pair of the filter $k$ and the spike train $spk$, either modeled or inferred spike trains. In the end, for each presynaptic neuron, we can evaluate the information carried by different spike trains for each filter. As a result, we can construct a MI matrix for different pairs of filters and inferred or modeled spike trains.

## White noise and natural image stimulus

Most of our analysis was conducted using white noise stimulus, as it is the preferred stimulus used for neuroscience experiments [17] and can be analyzed by STA ideally [29]. White noise images were generated as independent checkers of black pixel (-0.5) and white (0.5), similar to those used in experiments [17]. To test that STNMF has no restriction on the type of stimulus used to generate neural responses, we randomly selected 420000 images from ImageNet [35]. For each image, two image patches with 32x32 pixels were cropped to form a set of 840000 images, which were transformed into gray images and normalized to [-0.5 0.5] for all pixels. As a result, the magnitude of natural image intensity is similar to white noise while the texture shows rich natural scenes.

## Results

### Presynaptic spikes revealed by STNMF

The STNMF, inspired by spike-triggered analysis [29], was proposed to identify non-spiking bipolar cells from spike responses of RGCs [17]. Here we demonstrate that STNMF is able to reconstruct fully spiking neuron networks. For this, we created a model of the LGN neuron driven by four presynaptic RGCs, as shown in Fig 1A. The four presynaptic RGCs were modeled as linear-nonlinear Poisson neurons with different spatial receptive fields to compute local luminance. The LGN cell was modeled as an integrate-and-fire neuron receiving spikes of the four RGCs to produce spike trains in response to given sequences of stimuli (see Methods). Using visual stimuli consisting of a sequence of white noise as black and white checkers randomly distributed in space and time, the receptive field of such a neuron can be computed from spiking responses with reverse correlation or spike-triggered average (STA) [29]. However, the STA is equivalent to an average characteristic of the LGN cell as a combination of all RGCs in space, which can not provide any information about individual presynaptic RGCs.

Instead of averaging over spikes, the STNMF characterizes the spikes of the LGN cell as a nonlinear integration of all presynaptic RGCs, where each RGC computes stimulus in the first case. Thus, STNMF decomposes the LGN response using each output spike and input stimulus image as illustrated in Fig 1B. As a consequence of matrix factorization, we obtained a single weight matrix and a number of modules, where the number of modules is exactly the number of presynaptic cells in the model. The benefit of STNMF is that the modules are corresponding to the spatial receptive fields of upstream presynaptic cells [17], and the weight matrix encodes the information of synaptic connections [27]. In addition, the STNMF can separate all spikes of a postsynaptic neuron into different subsets of spikes for each presynaptic neuron, as illustrated in Fig 1C.

To illustrate the workflow of STNMF, we applied it to the model and analyzed the LGN spikes. Fig 1D(i)–1D(iii) shows the results of the STNMF analysis. It allows us to find four exact presynaptic RGCs with spatial and temporal filters modeled as in Fig 1Di. Using these filters, we recovered the nonlinearity component for each presynaptic RGC in Fig 1Dii, where different amplitudes are related to synaptic weights. The most notable feature of STNMF is that the weight matrix contains useful information about synapses. Two features can be extracted from the weight matrix, according to columns and rows, respectively. The first one is the synaptic weight from the RGC to the LGN cell in the model. To compute it, we averaged each column of the weight matrix to obtain the weight $W_j$ for each RGC $j$, which is exactly the synaptic weight from the presynaptic RGC, and matches the model component very well, even with different strengths between RGCs (Fig 1Dii). These results indicate that the STNMF weights provide a good estimate of actual synaptic connection weights from the RGCs to the LGN cell.

The second feature is based on the rows of the weight matrix. In the model, LGN spikes were contributed by four synaptic RGCs, thus, each spike of LGN cell could be triggered by one of the RGCs. We found that STNMF can classify all the LGN spikes into four subsets of spikes, where each subset of spikes is mainly, if not completely, contributed by one specific RGC. As each row corresponds to one individual spike, every spike can be attributed to one RGC as in Fig 1Diii. For this particular LGN model, we have four subsets of spikes for four RGCs respectively. To quantify the similarity between the RGC model spikes and STNMF inferred spikes, we computed pairwise cross-correlation to get a correlation matrix (Fig 1Diii, bottom left), which shows a good match between model spikes and STNMF inference. Interestingly, the correlation values of RGCs, the diagonal elements of the correlation matrix, are also positively correlated with synaptic weights. The RGCs with larger weights have a higher

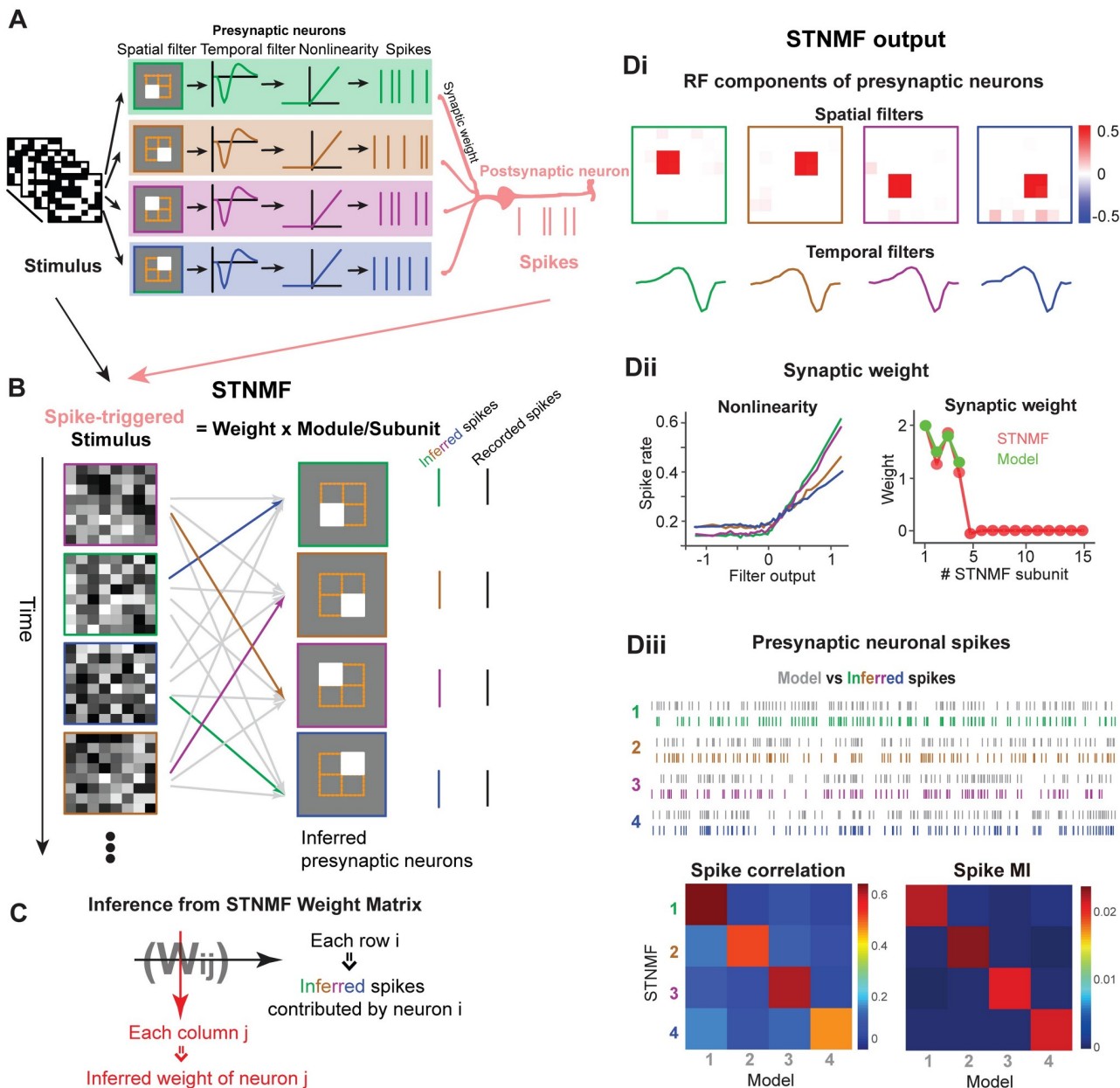

**Fig 1. Workflow of STNMF.** (A) Illustration of a minimal neural network model with four presynaptic RGCs and one postsynaptic LGN neuron. (B) Illustration of STNMF analysis. Averaging of the ensemble of spike-triggered stimulus images yields a single STA filter. STNMF reconstructs this ensemble by approximating it with a set of modules and a matrix of weights. One of the modules is strongly correlated to one of the spikes/images indicated by stronger (black lines) or weaker (gray lines) weights. (C) Illustration of STNMF weight analysis. Synaptic weights inferred by each column of the STNMF weight matrix, and spikes contributed by each presynaptic neuron inferred by each row of the matrix. (Di-iii) STNMF outputs. (i) Receptive field (RF) components of presynaptic neurons. Spatial filters (top) as subunit components of STNMF, and the corresponding temporal filters. (ii) Nonlinearity and synaptic weights from presynaptic neurons to the postsynaptic neuron. Ground-truth values of the model (green). The values computed from the weight matrix (red). Here weights were set as [2 1.5 1.8 1.3] for four neurons. (iii) STNMF separates the whole set of postsynaptic spikes into a subset of spikes contributed by each presynaptic neuron. Model spikes (gray) and inferred spikes (colored) of each presynaptic neuron. Correlation matrix of spike trains from model and STNMF (left). (Right) The matrix of mutual information (MI) carried by inferred spikes for each presynaptic neuron indicates that inferred spikes are similar to model spikes.

correlation for inferred spikes. We then computed mutual information carried by inferred spikes (Spike MI, Fig 1Diii, bottom, right) for each model RGC (see Methods). Higher values of MI along the diagonal line indicate that the inferred spikes of each RGC are more close to the target model RGC, but dissimilar to other non-target model RGCs. Mutual information gives similar results to quantify the spikes inferred by STNMF, thus through this study, we used the correlation between spikes as a measure in the results below.

These results suggest that the STNMF is feasible to dissect the network structure of spiking neural networks and allow us to obtain a complete set of functional components of the network related to spike responses. Particularly, spike trains inferred by STNMF are close to ground-truth spikes in the model.

## Inferring shared presynaptic neurons from multiple postsynaptic neurons

We then extended the LGN model to have multiple neurons in both layers, in which two LGNs share one part of visual space with overlapped receptive fields. The model was set up as follows (Fig 2): the first layer consists of six RGCs, each of which was modeled as previously with the identical temporal filter and nonlinearity, but spatial receptive fields are distributed at different locations of visual images. The output spikes are fed into two LGNs on the second layer, in which, the first LGN (L2–1) received spikes from RGC 1–4, and the second LGN (L2–2) received spikes from RGC 3–6. Thus, RGC 3–4 send information to both LGN cells.

Using a similar white noise input, we collected spikes of both LGN cells. Receptive fields of these two LGNs were obtained by STA as in Fig 2A. Leveraging the STNMF to both LGN cells, spatial receptive fields of each RGC (Fig 2A) are recovered as those in the model. Here we highlighted spikes inferred by STNMF for each RGC, which are similar to those in the model RGCs (Fig 2B). The similarity of spikes between model and inference was quantified by correlation coefficients as in Fig 2C, which shows that correlations are higher when paired with each own RGC. The same results were found for the LGN cell 2. Shared RGCs (No. 3 and 4) also show higher correlation values between inferred spikes from each individual LGN cell, even spikes were inferred by STNMF separately from each LGN cell. Taken together, these results imply that STNMF is able to reconstruct multiple spike trains within a network of multiple neurons.

## Inferring multilayered neurons using stimuli of white noise and natural images

Next, we extended the model to be three layers to simulate a neural circuit from RGC to LGN and V1. In this model, the first two layers are the same as that of the multi-LGN model, and in the third layer, a V1 neuron received spikes from both LGN cells (Fig 3A). The output spikes of the V1 neuron were collected for STNMF analysis. Using the white noise stimulus and STA to spikes of V1 neuron, the receptive field of the V1 neuron is an integration of receptive fields of six RGCs where shared RGCs show higher strengths. Whereas, we recovered all the spatial receptive fields of six RGCs using STNMF (Fig 3B). Surprisingly, when the STNMF was applied to the cell in the third layer, we actually recovered a set of the receptive fields of RGCs in the first layer directly. The number of captured cells converged and was independent of the subunit number assigned to STNMF (S1 Fig). It indicates that the nonlinear computation in the cascaded network perseveres information of stimulus from the input layer directly [17]. To verify that the STNMF captures the nonlinear computation of layer 1 cells, we reconstructed all the spike trains of six RGCs as in Fig 3C. The examination of the quality of RGC spikes was characterized by the correlation matrix between inferred and model spikes, with higher correlations for target RGCs (Fig 3C). To further examine this in layer 2, we pooled inferred spikes

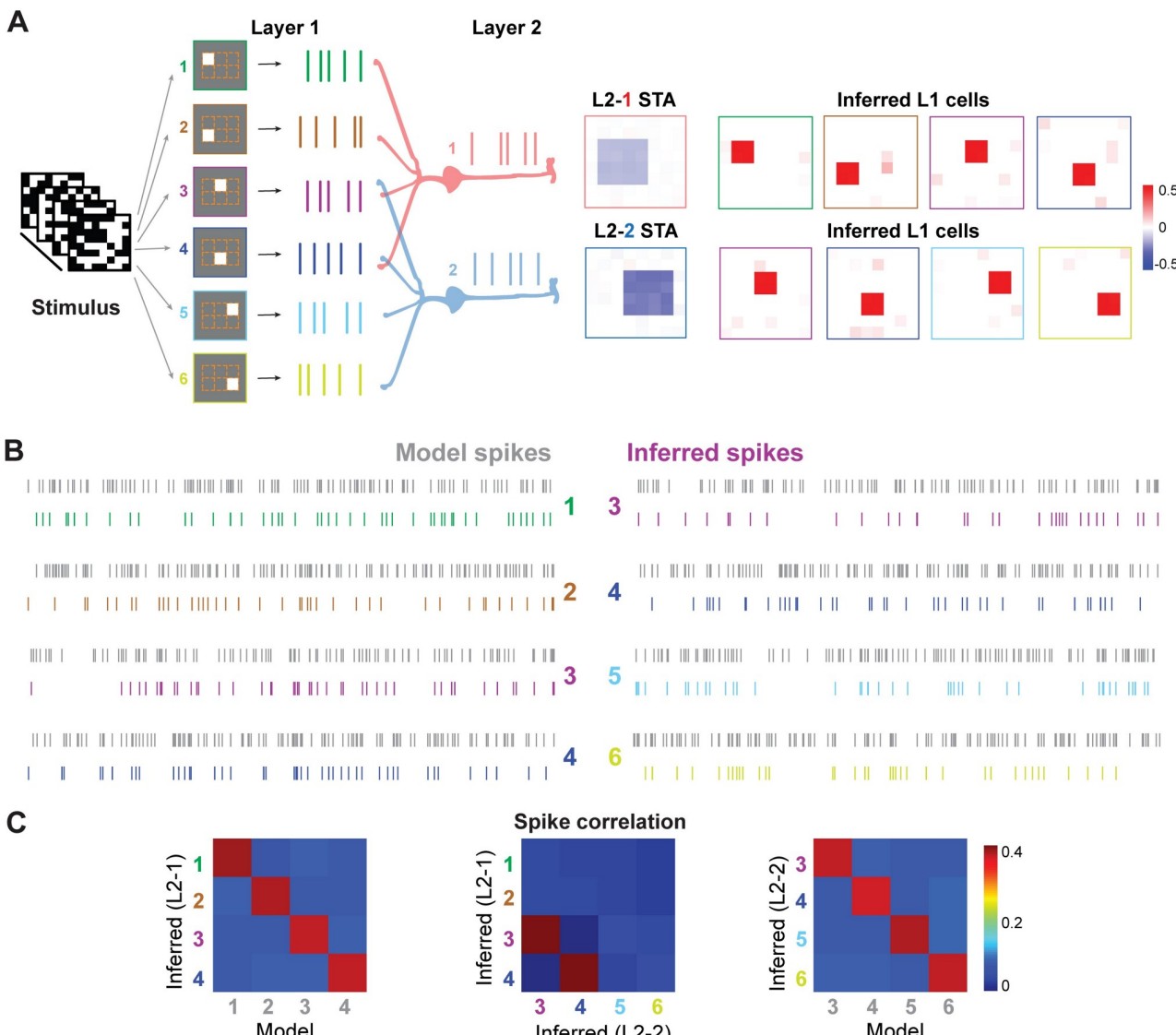

**Fig 2. STNMF inference of shared presynaptic cells from different outputs of postsynaptic cells.** (A) (Left) Illustration of a 2-layer network with two output neurons. Layer 1 (L1) has six cells, where the cell 3 & 4 target to both cells in layer 2 (L2). (Right) Inferred presynaptic cells independently from both cells (L2–1 and L2–2) of layer 2. Receptive fields computed by STA for cell L2–1 and L2–2 and inferred L1 cells. (B) Spike trains generated from layer 1 model cells (gray) and inferred by STNMF (colored). (C) Correlation matrices of spike trains for each cell of layer 1 computed between model cells and inferred spikes from L2–1 cell (left) and L2–2 cell (right), as well as between inferred spikes (middle).

of a set of four RGCs for individual LGN cells in layer 2. The LGN L2–1 pools over RGC 1–4, while LGN L2–2 over RGC 3–6, as in the model. Correlations between model and inferred spikes for layer 2 LGN cells show the similarity of spikes. Note here the self-correlation of the model and inferred spikes was included. However, the cross-correlation between the model and inferred spikes quantifies the similarity correctly.

We also considered various variations of the network model and found that STNMF is capable to infer a large number of cells (S2 Fig) and separate overlapped cells (S3 Fig), as well as works well on networks with more layers (S4 Fig). Furthermore, STNMF is still applicable to networks with weak recurrence (S5 Fig) and feedback (S6 Fig). Altogether, these results indicate that the STNMF works well for different scenarios of multilayer spiking networks, as

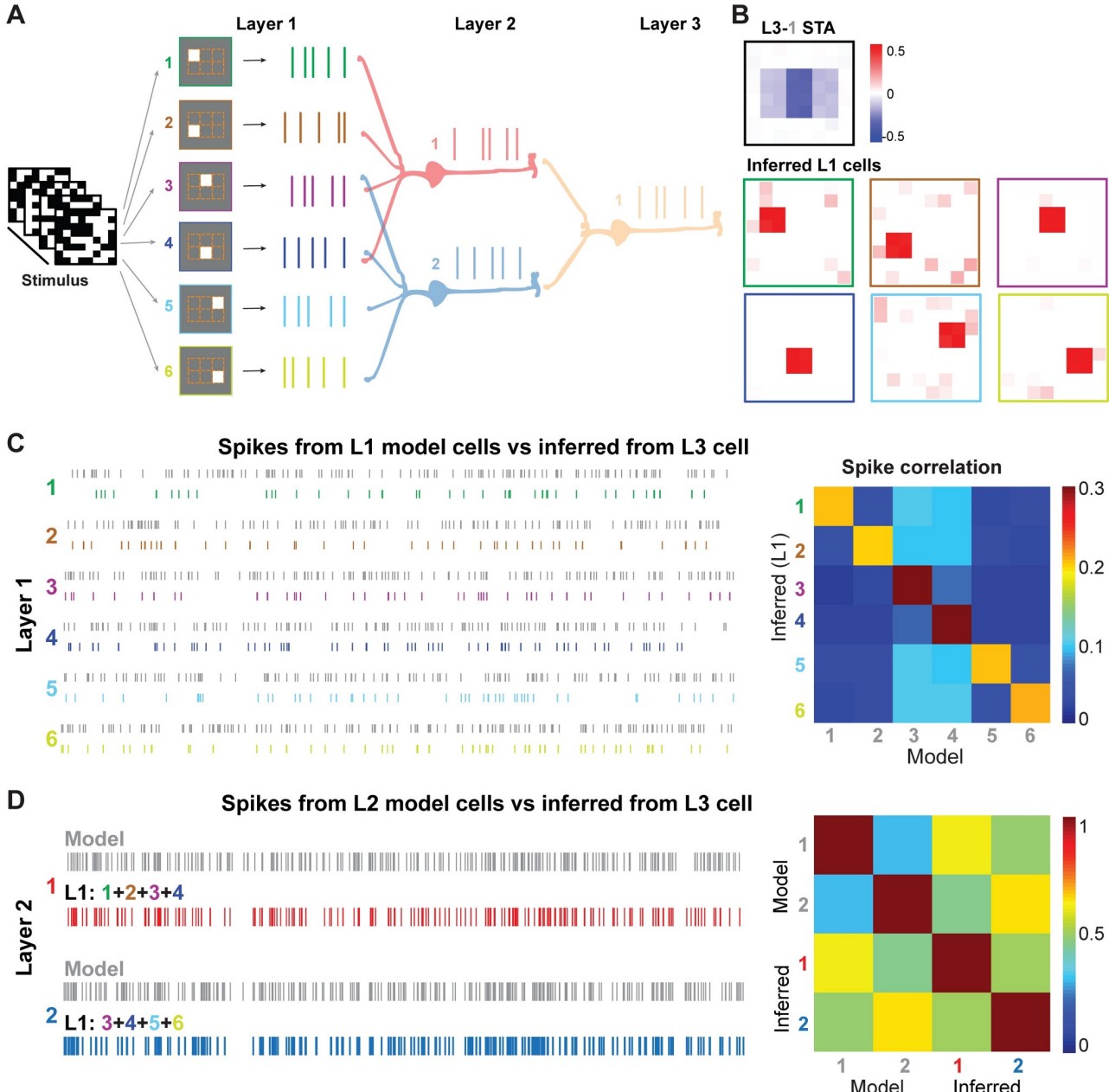

**Fig 3. STNMF analysis of a 3-layer model.** (A) Illustration of a 3-layer network. Layer 1 (L1) has six cells, where cell 3 & 4 target to both cells in layer 2 (L2). The layer 3 cell receives two layer 2 cells. STNMF was applied to the layer 3 cell. (B) Inferred presynaptic cells from the layer 3 cell. (Top) STA of cell L3–1. (Bottom) STNMF subunits are RFs of inferred L1 cells. (C) (Left) Spike trains generated from layer 1 model cells 1–6 (gray) and inferred from layer 3 cell by STNMF (colored). (Right) Matrices of corresponding correlation of spike trains between model and STNMF inference. (D) (Left) Spike trains generated from layer 2 model cells 1–2 (gray) and combined spikes inferred from layer 3 cell by STNMF (cell 1: red, cell 2: blue). (Right) Correlation matrices of spike trains between model and STNMF inference.

long as nonlinear computation, rather than linear computation, is manifested in neural networks.

To consider the generalization ability of STNMF to the complex stimulus images, we used a large set of natural images randomly selected from the ImageNet dataset [35] as stimuli in a 3-layer network (Fig 4). Unlike white noise stimulus, natural images make the STA analysis

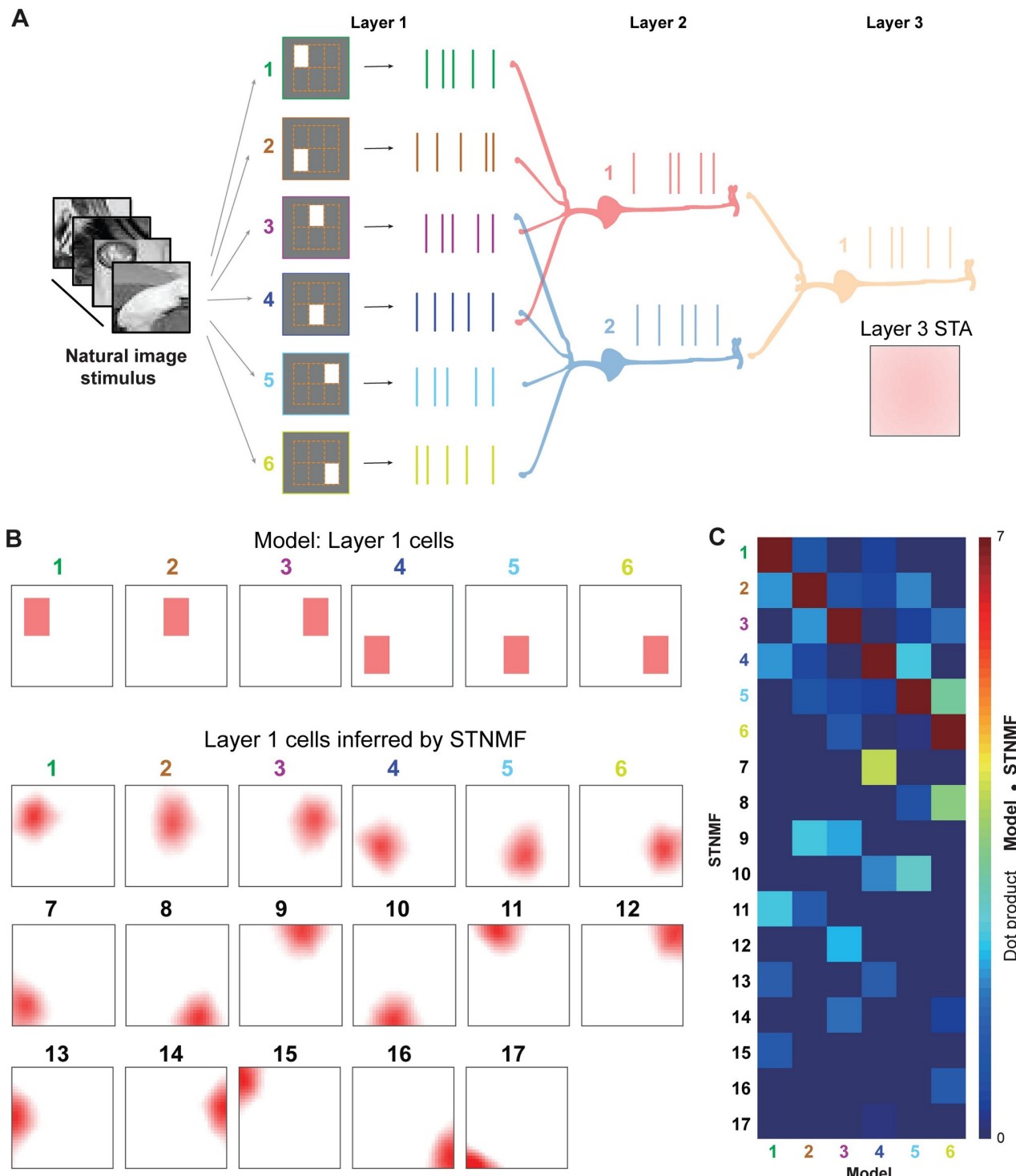

**Fig 4. STNMF analysis using natural image stimulus.** (A) Similar network model as in Fig 4 but with natural images instead of white noise as stimuli. The STA failed to obtain the RF of Layer 3 cell. (B) Modeled RFs of Layer cells (top) and the STNMF inferred results with 17 subunits (bottom). (C) Dot product matrix of the model RFs and STNMF subunits showing that the first subunits resemble the model cells.

failed to get the RF. Indeed, using a large set of natural images, the RF of the Layer 3 cell can not be obtained by STA (Fig 4A). In contrast, the STNMF is capable to infer all the RFs of Layer 1 cells in the model (Fig 4B). The quantitative metric using the dot product between the RFs of model cells and the subunits inferred by STNMF confirms that the model RFs are highly overlapped with the STNMF inferred results (Fig 4C). These results show that STNMF enables us to disentangle the nonlinear computation of the network under complex natural images in an interpretable way.

## Inferring simple and complex cells

So far we considered the model with the same type of cells as OFF cells. There are different cell types in neural systems. In the retina, there are at least two functionally distinct cell types, ON and OFF cells, where ON cells are more sensitive to light increments resulting in an opposite sign for the receptive field, whereas OFF cells are functionally opposite for light decrements. To examine the feasibility of STNMF for a network with mixed cell types, we designed a network with both ON and OFF cells showing different polarities of the receptive field as in Fig 5A. As the entire receptive field filter is a multiplication of spatial and temporal filters, we fixed the temporal filter as negative, while flipped the spatial filter as positive for ON cells.

Following the steps as above for the OFF-cell network, the STNMF was applied to the output spikes of the layer 2 cell, which received inputs from both ON and OFF RGCs. We found STNMF can retrieve individual presynaptic ON and OFF components, whereas the receptive field of layer 2 cell computed by STA shows a mixture of ON and OFF features. Since the spatial filters as STNMF modules are always positive, the corresponding temporal filters show different polarities according to the ON and OF cell types (Fig 5B). Consequently, the spikes associated with each presynaptic neuron were extracted using the maximal values of the weight matrix for each spike (see Methods). In this way, we have a set of spikes for all presynaptic neurons, yet maintaining cell types. To justify the cell types, we computed the receptive field of each RGC applying the standard STA to inferred RGC spikes. The obtained receptive fields of each RGC in Fig 5C show typical ON and OFF features. The sum of weight values, either the specific values of each RGC, or all weights in the STNMF weight matrix (Fig 5D) confirms that ON and OFF cells can be determined by weight values as the minimums for OFF and the maximums for ON cells. The quality of inferred spikes for each RGC was characterized well by the correlation of spike trains between model and inference.

Finally, we simulated V1-like simple and complex cells using a 3-layer network, where there are both ON and OFF RGCs in the first layer receiving a mixture of light information as in Fig 6A. The V1 simple cell at layer 3 has a typical receptive field with mixed ON and OFF features (Fig 6B). STNMF was applied to the layer 3 cell to retrieve a set of subunits, which resemble the layer 1 ON and OFF cells, where the polarities of ON and OFF type are indicated by the signs of the peaks in temporal RF filters (Fig 6B). The spikes of the V1 cell were decomposed into a set of spikes as in Fig 6C, each of which is closely associated with the layer 1 RGC spikes, assessed by the correlation between spike trains. The spikes of layer 2 LGN cells in Fig 6D were achieved by pooling the spikes of corresponding layer 1 RGCs. These results indicate that we can utilize STNMF for V1-like simple cells to decouple the mixture of cell types in the network.

To model V1-like complex cells, we used a similar 3-layer network with two sets of layer 1 RGC cells Fig 7A. Each set of layer 1 cells was distributed in four spatial locations, however, both sets have the opposite polarity of ON and OFF receptive fields, resulting in two different layer 2 LGN cells located at the same space but opposite RFs. As a result, the layer 3 cell is similar to a V1 complex cell, for which the standard STA analysis fails to generate the RF (Fig 7A).

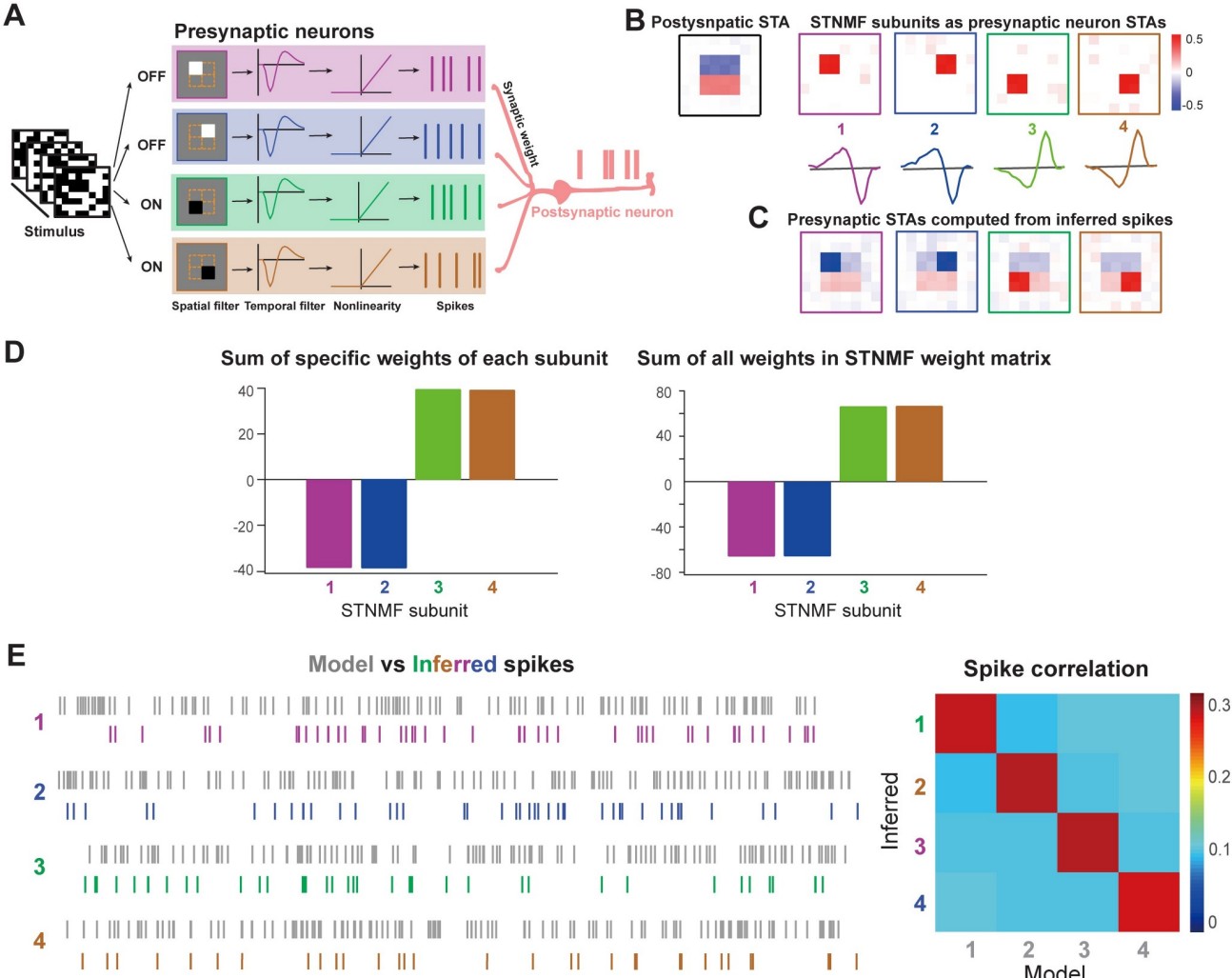

**Fig 5. Mixture of ON and OFF cell types identified by STNMF.** (A) Illustration of a neural network with ON and OFF cells. Similar to Fig 1, except that there are both ON and OFF presynaptic neurons. (B) ON and OFF cells are separated from STNMF. The RF of the postsynaptic neuron computed by STA (left). RFs of presynaptic neurons identified as STNMF subunits (top) with their corresponding temporal filters (bottom). (C) Presynaptic RFs computed by spikes inferred from STNMF. (D) Using STNMF weight matrix to classify spikes, the relationship among spikes, weights, and subunits is established, seen from (left) sum of specific weights of each subunit, and (right) sum all weights in each column of the weight matrix. (E) Spikes from the model and inferred by STNMF (left), and the corresponding matrices of spike correlation.

Remarkably, the analysis of STNMF can retrieve a set of subunits resembling layer 1 cells. When using eight modules in STNMF, we found there are four subunits converging to layer 1 cells, while other subunits are noise (Fig 7B). Due to the nature of the non-negative analysis, the subunits resulting from STNMF are always positive, thus, these four meaningful subunits represent all layer 1 cells.

To further separate ON and OFF layer 1 cells, the spikes of the V1 complex cell were extracted using each subunit. For each subunit, the set of spikes was obtained by the minimal values of the STNMF weight matrix for OFF cells, whereas spikes by the maximal values for ON cells as in Fig 7C. Therefore, we recovered eight spike trains from four subunits, and the correlation matrix of spikes shows that they are highly linked to layer 1 cells. To assure that these spikes are meaningful, we used the standard STA analysis for each set of spikes and

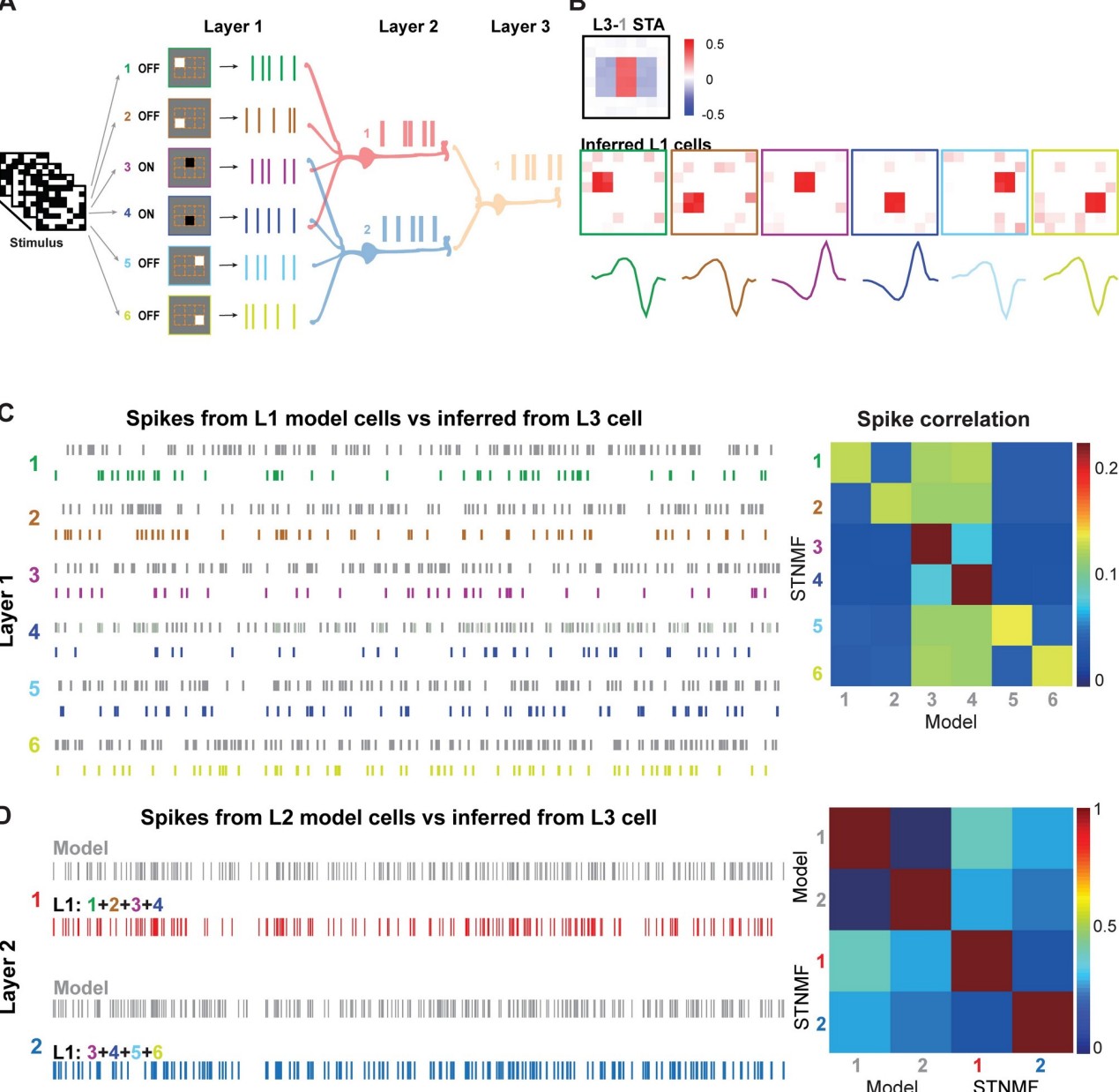

**Fig 6. STNMF analysis of a V1 simple cell.** (A) Illustration of a model simple cell as a 3-layer neural network with ON and OFF cells. (B) ON and OFF cells are separated from STNMF. The RF of the simple cell computed by STA. RFs of layer 1 cells were identified as STNMF subunits with their corresponding temporal filters. (C) (Left) Spike trains generated from layer 1 model cells 1–6 (gray) and inferred from layer 3 cell by STNMF (colored). (Right) Matrices of corresponding correlation of spike trains between model and STNMF inference. (D) (Left) Spike trains generated from layer 2 model cells 1–2 (gray) and combined spikes inferred from layer 3 cell by STNMF (cell 1: red, cell 2: blue). (Right) correlation matrices of spike trains between model and STNMF inference.

obtained the spatial and temporal filters as in Fig 7D. Both the spatial and temporal filters are similar to the model cells of layer 1, and the polarity of spatial filters determines the ON and OFF cells. Altogether, these results demonstrate that the STNMF is applicable to not only the cells in the retina but also in the LGN and V1 parts of the visual system. The intricacy of nonlinear complex cells in V1 can also be unfolded by STNMF.

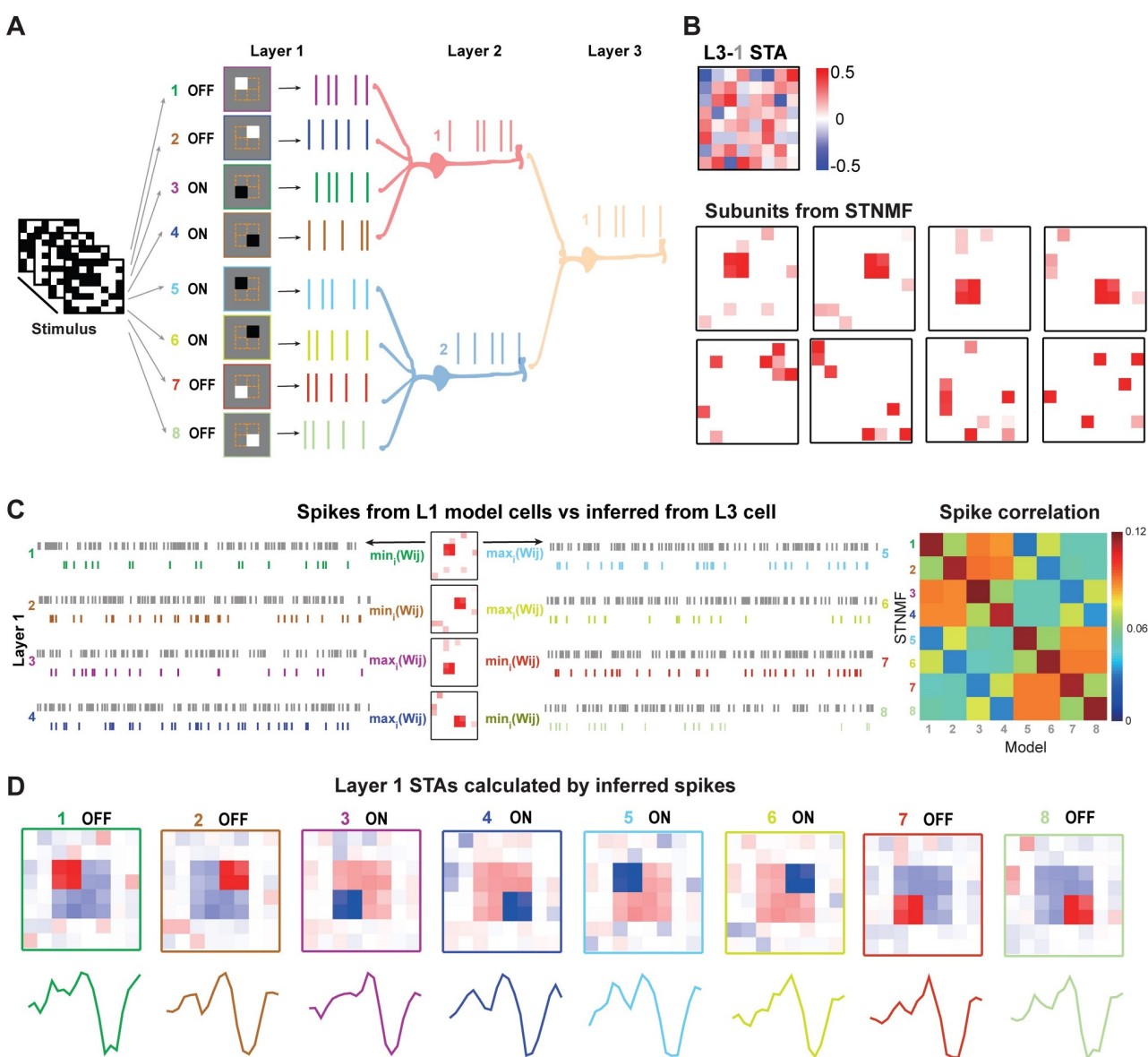

**Fig 7. STNMF analysis of a V1 complex cell.** (A) Illustration of a model complex cell as a 3-layer neural network with ON and OFF cells. There are eight cells in layer 1, of which the first four cells 1–4 have mixed ON and OFF types, and the second four cells 5–8 are similar with the same locations but with opposite polarity. Layer 2 cells are simple cells as in Fig 5. Layer 3 cell is a V1 complex cell. (B) The RF of the complex cell calculated by STA. 8 Subunits obtained by STNMF. (C) (Left) Spike trains generated from layer 1 model cells 1–8 (gray) and inferred from layer 3 cell by STNMF (colored). For each meaningful subunit, spikes are separated by minimal values of the weight matrix as OFF spikes, and minimal as ON spikes, respectively. Totally, there are 8 classified spike trains. (Right) Matrices of corresponding correlation of spike trains between model and STNMF inference. (D) Spatial and temporal filters obtained by STA analysis using classified spikes.

## Discussion

In this study, we demonstrated that the STNMF is capable to dissect functional components of spiking neural networks and reconstruct spike trains of presynaptic neurons by analyzing spikes of the output neurons. Within feedforward networks with multiple stages or layers and multiple neurons, applying STNMF to spikes of neurons at the final layer allows us to recover the entire neural network, not only the structural components of neurons and synapses, but neuronal spikes of cascaded layers, which transfer the input stimulus to final output neurons.

These results suggest the STNMF is a useful technique for interpreting neural spikes and uncovering the relevant functional and structural components of neuronal circuits.

## 0.1 The role of presynaptic neurons in postsynaptic neural spikes

Here we demonstrated the scenarios where a postsynaptic neuron receives a few presynaptic neurons that all are firing spikes and contribute to the firing of the postsynaptic neuron. It is well known that a neuron has morphology with a dendritic tree receiving nonlinear inputs from presynaptic neurons [36]. The neuronal morphology varies significantly, depending on the cell type, location, and brain area [37]. Similarly, the firing rate of cells also varies remarkably depending on the dynamic states of synaptic strength [38]. A typical cortex cell with thousands of synapses maintains a very low firing rate [39].

Recent evidence, using advanced experimental techniques recording the activity of single synapses *in vivo*, shows that single synapses could be active, while the population of synapses is rather spare [40]. This indicates that, in terms of spikes of a postsynaptic neuron, only a small subset of synapses actively contribute to the somatic firing at one time, while most of the synapses are silent. Experimental observations and theories utilizing this feature suggest complex scenarios of the interaction of spare synaptic firing and dendritic computation at the single-cell level [40], and spare neural coding at the level of neural circuits [41]. The STNMF may have an advantage in utilizing these shreds of evidence for understanding the computational principle of neural coding.

## 0.2 Reconstruction of the dynamics of neuronal networks

Recent experimental advances provide tools to reconstruct large-scale neural circuits [8, 9] and relatively stereotyped retinal circuits [1, 3, 4]. However, these static connectomic structures can not explain ever-changing neuronal dynamics and reveal valuable functions performed by neurons. Taking the example of direction selectivity in retinal neurons, the structure basis was suggested as the asymmetrical distribution of inhibitory amacrine cells around ganglion cells [42], however, direction selectivity is rather dynamical and reversible [43]. Thus it is important to reconstruct the functional dynamics of neural networks.

The methods that can analyze network connectivity using neural response are still limited. Granger causality [44], dynamic causal modeling [45], and transfer entropy [46]) are popular methods used for this purpose yet with certain limitations [47–50]. The STNMF, as a relatively new method, provides a different means to systematically investigate functional neural circuits using spikes. Together with other recent studies focusing on dynamical structures of neural networks [16, 18], it is possible to incorporate dynamic components, such as synaptic strengths and presynaptic spikes, to reveal detailed functional organization of neural circuits.

## 0.3 Inferring neural spikes

The complexity of dendritic organization in neurons depends on the type of neurons. For some neurons, such as Purkinje cells in the cerebellum, there is a large dendritic tree receiving tens of thousands of presynaptic inputs [51]. However, some neurons, such as unipolar cells of the cerebellum, have only one dendrite receiving one presynaptic input [52]. Yet, the underlying computations in both types of neurons are rich [53]. It is thought that many synapses are silent, perhaps at particular time points, during the spike dynamics of postsynaptic neurons. Thus, it is meaningful to extract the contribution of presynaptic neurons from the viewpoint of postsynaptic neurons. Here we noticed that STNMF can classify spikes of postsynaptic neurons into a set of spikes, where each set is considered as ab overall contribution of presynaptic neurons.

The results of STNMF are meaningful, in that it allows us to obtain the dynamic strengths of presynaptic cells, according to whether they deliver the effect on spike dynamics of postsynaptic cells. Therefore, the outcome of STNMF is naturally for inferring spikes to capture the underlying dynamics of neural circuits, rather than static connections between neurons. In this sense, STNMF could provide more information than Granger causality, which tells the direction of information between neurons [54].

For experimental data where no spike can be exacted, such as graded signals in retinal bipolar cells [55, 56], or the coarse version of neural signals, such as neuronal calcium imaging data [15, 26], and local field potentials representing a small or large network of neural population [57, 58], STNMF could be potentially applied to extract useful information within neural circuits, as long as neural signals are dynamics with meaningful states reflecting neural spikes. Given recent advances in experimental techniques for simultaneously recording multiple brain areas with single cell resolution [59], these data could yield interesting protocols for utilizing STNMF on the level of large scale neural circuits.

## 0.4 Multilayered neuronal networks

A ubiquitous feature of neural circuits in the brain is that neurons are organized by layers or stages. Although there are dramatic feedback and/or recurrent connections between neurons [60], the information flow within recurrent neural networks could reinforce neurons to form a prevailing feedforward format of dynamics, utilizing synaptic plasticities [61, 62]. One prominent example is the neural trajectory, in which different neurons fire at particular time points so that the overall dynamics of neural populations becomes a trajectory spanning over time, such as songbird neural dynamics [63], and space, such as memory dynamics of place cells [64].

Nevertheless, the dynamics of the neural network is controlled by multiple layers and pathways [65, 66]. In some neural systems, feedforward networks are more prominent. The typical example is the visual pathway modeled here, starting the retina to LGN and visual cortex. The relatively simple organization of the retinal circuit makes it a perfect system for dissecting the dynamics and computations of the the multilayered neural network [67, 68]. Leveraging the feature of macaque retina with less dense distribution and large size of photoreceptors away from the fovea, the STA analysis, using fine-size white noise checker images, can infer photoreceptors of the input layer while analyzing the spikes of ganglion cells of the output layer [67]. However, such an approach is difficult for analysis of general retinal neural systems, and STA analysis can not detect bipolar cells of the hidden layer [17]. The STNMF was introduced to consider the restricted two-layer network of bipolar cells and ganglion cells, where there are no spikes in bipolar cells [17, 27]. Here we demonstrated that STNMF is applicable to fully spiking neural networks with multiple layers. It is well known that a simple three-layer perceptron with one hidden layer can greatly expand the computational power of artificial neural networks. Similarly, multilayered neural network presents many interesting features, such as synfire chains [69], of neural activities in neuroscience, resembling some experimental observations, such as songbird neural dynamics [63]. STNMF could serve as a tool for understanding these dynamics.

Much effort has been made to characterize the neuronal receptive fields in LGN and visual cortex [70–74]. However, the computation in the visual pathway is carried out by different layers and stages [65, 75], and there is no efficient way to dissect them systematically across multiple layers [76]. Here we demonstrated that the STNMF is able to identify the receptive fields of neurons in the input layer, even the STNMF was applied to output neurons in the final layer.

Such an across-layer analysis of STNMF is a manifestation of nonlinear computation within neuronal networks. Spike response of neurons is an indication of the nonlinear computation using various ion channels in neurons [52]. Thus, the STNMF, leveraging the advantage of NMF for describing local structures of images, can naturally fit in the neuronal systems with spikes.

The ultimate goal of reconstructing neural circuits is to utilize those neural and synaptic components for neural computation. In recent years, detailed neuroscience knowledge strengthens the bottom-up approach of neural network modelling [77], in which one prominent feature is to utilize neuroscience-revealed network structures to design, rather than handcraft, possible artificial network architectures [78]. Here we indicated that the STNMF can detect computational components across layers or stages of cascade neural networks. Recent studies show that NMF variants can be combined with the framework of multilayer architecture [79, 80] to learn a hierarchy of attributes between layers. Thus, one future direction is to extend STNMF to infer all the computational components simultaneously in multilayered neural networks. Therefore, further extension of STNMF is likely to be fruitful for understanding the hierarchical architecture of neuronal systems in the brain.

## 0.5 Limitations

A variety of advanced experimental techniques in neuroscience can measure different types of functional neural signals. Spiking signal is one of the many formats. Other continuous signals measured for single cells, such as two-photon calcium imaging, as well as for coarse-scale cell ensembles, such as electroencephalogram and functional magnetic resonance imaging, can not infer spikes directly. Further effort is needed to adapt STNMF to investigate these non-spiking signals. Meaningful neural responses are often defined as peaks of these signals. Recent studies imply there is a close correlation between peaks of a neural signal of two-photon calcium imaging with spikes [81, 82]. Thus, extracting peaks as spikes can make STNMF work for neural calcium imaging signals. Systematic studies are deserved for detailed examination of the coarse-scale non-spiking neural signal using STNMF.

Although neural circuits are organized by layers across the brain and sensory information flows in a feedforward way, recurrent connections between neurons are also prevailed and useful for dynamic coding [83, 84]. We showed that STNMF can work well in networks with weak recurrence and feedback. Future work is needed to extend STNMF to take into account recurrence. However, these structure indices are rather static. Dynamical routing of information in a network is more dramatic, which makes networks be in a regime of feedforward dynamics with recurrent structures [61]. Recent studies using graph theory suggest that neural network in the brain contains multiple ensembles of local community or module subnetworks [85]. One possible way is to utilize the coding principle of sparse firing and ensemble firing in a large network to separate the whole network into a set of local networks. One can apply STNMF iteratively and hierarchically through subsets of local networks for disentangling the effect of recurrent and feedforward connections on the information flow.

## Supporting information

**S1 Fig. Related to Fig 3. The inferred results of STNMF converge to the right number of presynaptic neurons if K is set to be a larger value.** The receptive fields of extra modules are noisy.
(TIF)

**S2 Fig. Related to Fig 3. STNMF is able to infer more cells across layers in a network.** The three-layer network has 16 cells in Layer 1. Among them, neurons 1–8 are connected to the first neuron of Layer 2, and neurons 9–16 are connected to the second neuron of Layer 2. The STA shows the receptive field of the Layer 3 cell. The receptive fields of modeled Layer 1 cells are recovered by the STNMF inference.
(TIF)

**S3 Fig. Related to Fig 3. STNMF enables to infer presynaptic cells with overlapped receptive fields.** There are four Layer 1 cells with overlapped receptive fields. The STA shows the overall receptive field of the Layer 3 cell, while the STNMF separates them into individual ones of Layer 1 cells.
(TIF)

**S4 Fig. Related to Fig 3. STNMF inference in a four-layer network.** Similar to Fig 3 but with a four-layer structure. The STA shows the receptive field of the Layer 4 cell, while the STNMF obtains the receptive fields of Layer 1 cells.
(TIF)

**S5 Fig. Related to Fig 3. STNMF is able to infer cells in the network with weak recurrence.** Similar to Fig 3 but with a recurrent connection from the Layer 3 cell to cell 2 in Layer 2. The recurrent connection weight is 0.1, compared to other weights as 1. STNMF can infer the receptive fields of Layer 1 cells.
(TIF)

**S6 Fig. Related to Fig 3. STNMF is able to infer cells in the network with weak feedback.** Similar to Fig 3 but with a feedback connection from the Layer 3 cell to cell 1 in Layer 2. The feedback connection weight is 0.1, compared to other weights as 1. STNMF can infer the receptive fields of Layer 1 cells.
(TIF)

## Author Contributions

**Conceptualization:** Zhaofei Yu, Jian K. Liu.

**Data curation:** Shanshan Jia, Jian K. Liu.

**Formal analysis:** Shanshan Jia, Zhaofei Yu, Jian K. Liu.

**Funding acquisition:** Dajun Xing, Zhaofei Yu, Jian K. Liu.

**Investigation:** Shanshan Jia, Dajun Xing, Zhaofei Yu, Jian K. Liu.

**Methodology:** Shanshan Jia, Zhaofei Yu, Jian K. Liu.

**Project administration:** Zhaofei Yu, Jian K. Liu.

**Resources:** Shanshan Jia, Jian K. Liu.

**Software:** Shanshan Jia, Jian K. Liu.

**Supervision:** Zhaofei Yu, Jian K. Liu.

**Validation:** Shanshan Jia, Dajun Xing, Zhaofei Yu, Jian K. Liu.

**Visualization:** Shanshan Jia, Dajun Xing, Zhaofei Yu, Jian K. Liu.

**Writing – original draft:** Shanshan Jia, Dajun Xing, Zhaofei Yu, Jian K. Liu.

**Writing – review & editing:** Shanshan Jia, Dajun Xing, Zhaofei Yu, Jian K. Liu.

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
