## [Decision Letter · Decision Letter 0]

20 Aug 2021

Dear Dr Yu,

Thank you very much for submitting your manuscript "Dissecting cascade computational components in spiking neural networks" for consideration at PLOS Computational Biology.

As with all papers reviewed by the journal, your manuscript was reviewed by members of the editorial board and by several independent reviewers. In light of the reviews (below this email), we would like to invite the resubmission of a significantly-revised version that takes into account the reviewers' comments.

We cannot make any decision about publication until we have seen the revised manuscript and your response to the reviewers' comments. Your revised manuscript is also likely to be sent to reviewers for further evaluation.

Sincerely,

Tianming Yang

Associate Editor

PLOS Computational Biology

Lyle Graham

Deputy Editor

PLOS Computational Biology

Reviewer's Responses to Questions

**Comments to the Authors:**

Reviewer #1: In the manuscript entitled “dissecting cascade computational components in spiking neural networks”, Jia et. al. exploited a data analysis method named spike-triggered non-negative matrix factorization (STNMF) to infer the dynamical structure of cascade neural networks. The effectiveness of this method has been successfully justified by using the simulation data of multi-layer neural networks in the early visual pathway. As demonstrated by the examples in the manuscript, STNMF is quite effective in extracting the contribution of presynaptic neurons to each spike of a postsynaptic neuron and recovering the connectivity strengths, which will be potentially useful to understand the information coding of feedforward-type neural circuits. The manuscript is already well organized and clearly written, so I only have a few minor comments as listed below.

1. A potential practical problem of STNMF is that one needs to have some prior knowledge about the number of modules/neurons in the first layer, i.e., the parameter K in page 5. But this information is usually hard to obtain. So, is it possible to propose some model selection criteria to help choose an optimal P when applying STNMF?

2. It seems that STNMF may be less effective if neurons in the first layer have spatially overlapped receptive fields. Although this case is not fully investigated in the manuscript, some indications can be seen from Fig. 3C right panel. It will be helpful if the authors could discuss a bit on this more realistic scenario.

3. When inferring the contribution of presynaptic neurons to each spike of a postsynaptic neuron, the corresponding biological interpretation may require more discussion. In general, a resultant spike of a postsynaptic neuron shall attribute to input spikes from multiple presynaptic neurons, not necessarily the case that only one of the presynaptic neurons contributes dominantly. It seems that, when the presynaptic neurons’ firing rate is low and the synaptic strength is relatively large, then the inferred spike will be dominantly influenced by a single presynaptic neuron that fires latest. Is the modeling network in this regime?

4. Line 92: The math notation r=f(k^Ts(t)) may be misleading. It’s better to change it to r=f(k*s(t)), where * represents spatiotemporal convolution.

5. Line 104: W_i shall be w_i.

6. Line 106: To be precise, it is better to add a Heaviside function in the expression of the synaptic current, indicating that t starts from t_i^j, and before that time one always has I_syn=0.

7. Line 135: \\bar{s}^I shall be defined as either the integral or the summation of the right-hand-side term over \\tau.

8. Line 145: Why sparsity is constrained on each column of M rather than each row of M? It is deemed that each row of M is the receptive field of a presynaptic neuron, which needs to be sparsified (because the size of a receptive field is small).

9. Line 154, “LNG” shall be “LGN”.

10. Line 204, “80100%” shall be a typo.

Overall, it is deemed that STNMF is an effective method to infer the dynamical structure of cascade neural networks. In contrast to Granger causality (a popular data analysis method in neuroscience research), STNMF can give more information of the dynamical interaction between neurons. In addition, STNMF may be more applicable to nonlinear neural network dynamics than Granger causality, as Granger causality is proposed based on linear regression models so it can lead to incorrect inference results for neuroscience data, as shown in previous studies:

Stokes P, Purdon P. A study of problems encountered in Granger causality analysis from a neuroscience perspective. PNAS, 114 (34) 7063-7072, 2017.

Li S, Xiao Y, Zhou D, Cai D. Causal inference in nonlinear systems: Granger causality versus time-delayed mutual information. Physical Review E, 97, 052216, 2018.

Reviewer #2: The manuscript "Dissecting cascade computational components in spiking neural networks" by Jia et al.,

has employed a recent method, termed spike-triggered non-negative matrix factorization (STNMF) to

extract the functional connections in a set of simulated two- and three-layer feedforward network models.

The results proved that STNMF method could be able to successfully derive the connectivity structure

and provide predictions of synaptic weights. The results are solid and interesting and I like the STNMF method,

while there are several concerns/flaws make it hard to be convinced for the PLoS Comp Biol. journal.

1. strictly speaking, the STNMF method was developed in authors' previous publication (ref.17), and already was

applied in a two-layer feedforward network model (ref.27), hence the present manuscript (with add results of 3-layer

network) does not contain enough novelties. Also, there are a lot of repeated results/statements were already

done in previous works.

2. second, although authors claimed several times in the manuscript that the STNMF method could be a general

approach for neural systems with kinds of functional connectivity structures. However, it

may be overstated since this paper as well as their previous works only show that it works well for

feedforward-type of networks, but no evidence for networks with

feedback or other re-current connectivities.

3. indeed if only apply the method for the simulated network models, authors could have more open directions

to extend the STNMF method, for example, this manuscript could consider over the white noise-type stimuli,

since previous work by Jack Gallant group showed that colored noise-type stimuli could be better one to detect

the neurons' receptive field.

minor points:

in several places, LGN was mistyped by LNG

Reviewer #3: The authors applied non-negative matrix factorization (NMF) to the analysis of neuronal networks. They tested the method on model circuits that are similar to the early visual systems by inferring information regarding the circuit using the spike trains of postsynaptic neurons.

The core idea is that the stimuli (movies) are first convolved with the temporal STAs at each pixel. These convolved images at the time of spikes were subjected to NMF to give two matrices representing the effective inputs from individual modules and the spatial patterns of those modules. The authors showed that the method successfully estimated the stimulus-response properties of the presynaptic neurons and the synaptic weights for network models including three-layer models.

The method is interesting and would be useful for some kinds of network analyses. For the paper to be fully useful for the neuroscience community, I recommend improving the following points.

(1) Determination of the number of modules

For NMF to be successful, determining the number of modules (K) is critical. If my understanding is correct, authors appear to assume that K is given. However, this is unlikely in the analysis of real neuronal responses. Please discuss this point or include a method to estimate K.

(2) Overlap of spatial filters.

In all the circuit models shown in the paper, the spatial filters of RGCs with the same sign (ON or OFF) do not overlap. This is unlikely in real neuronal circuits. In addition, because the method uses pixel-wise temporal convolution to generate the matrix S, the fact that the spatial filters do not overlap may have contributed to the results. Please discuss whether the method works when RGCs with the same sign spatially overlap.

(3) The number and density of presynaptic neurons

The models in the paper contain relatively small number (≦10) of presynaptic neurons. Although this is reasonable for method evaluation and for certain neuronal circuits, many neurons receive non-trivial inputs from a much larger number of presynaptic neurons. Please discuss whether the method works for such networks.

Below are minor points:

Line 104: “Wi” should be in lowercase.

The manuscript appears to contain non-standard mathematical notations. For example,

Line 126: In the notation for the weighted average, a dot seems to represent the dot product, but this is not clearly stated.

Line 138: The suffix i appears unnecessary.

Please carefully check mathematical notations.

**Have the authors made all data and (if applicable) computational code underlying the findings in their manuscript fully available?**

Reviewer #1: Yes

Reviewer #2: **No: **it will be better if authors provide data samples and computational codes for audients to reproduce their figures.

Reviewer #3: Yes

PLOS authors have the option to publish the peer review history of their article (what does this mean?). If published, this will include your full peer review and any attached files.

Reviewer #1: No

Reviewer #2: No

Reviewer #3: No
---

## [Decision Letter · Decision Letter 1]

14 Nov 2021

Dear Dr Yu,

We are pleased to inform you that your manuscript 'Dissecting cascade computational components in spiking neural networks' has been provisionally accepted for publication in PLOS Computational Biology.

Best regards,

Tianming Yang

Associate Editor

PLOS Computational Biology

Lyle Graham

Deputy Editor

PLOS Computational Biology

Reviewer's Responses to Questions

**Comments to the Authors:**

Reviewer #1: The authors have addressed all my questions.

Reviewer #2: well done

**Have the authors made all data and (if applicable) computational code underlying the findings in their manuscript fully available?**

Reviewer #1: Yes

Reviewer #2: Yes

PLOS authors have the option to publish the peer review history of their article (what does this mean?). If published, this will include your full peer review and any attached files.

Reviewer #1: No

Reviewer #2: No

---

## [Editor Report · Acceptance letter]

22 Nov 2021

PCOMPBIOL-D-21-01074R1 

Dissecting cascade computational components in spiking neural networks

Dear Dr Yu,

I am pleased to inform you that your manuscript has been formally accepted for publication in PLOS Computational Biology. Your manuscript is now with our production department and you will be notified of the publication date in due course.

With kind regards,

Zsofia Freund
